There are amendments to this paper

# De novo identification of essential protein domains from CRISPR-Cas9 tiling-sgRNA knockout screens

Wei He[1,5], Liang Zhang[1,5], Oscar D. Villarreal[1], Rongjie Fu[1], Ella Bedford[1], Jingzhuang Dou[1], Anish Y. Patel [1], Mark T. Bedford [1,2], Xiaobing Shi [3], Taiping Chen [1,2], Blaine Bartholomew [1,2] & Han Xu[1,2,4]*

High-throughput CRISPR-Cas9 knockout screens using a tiling-sgRNA design permit in situ evaluation of protein domain function. Here, to facilitate de novo identification of essential protein domains from such screens, we propose ProTiler, a computational method for the robust mapping of CRISPR knockout hyper-sensitive (CKHS) regions, which refer to the protein regions associated with a strong sgRNA dropout effect in the screens. Applied to a published CRISPR tiling screen dataset, ProTiler identifies 175 CKHS regions in 83 proteins. Of these CKHS regions, more than 80% overlap with annotated Pfam domains, including all of the 15 known drug targets in the dataset. ProTiler also reveals unannotated essential domains, including the N-terminus of the SWI/SNF subunit SMARCB1, which is validated experimentally. Surprisingly, the CKHS regions are negatively correlated with phosphorylation and acetylation sites, suggesting that protein domains and post-translational modification sites have distinct sensitivities to CRISPR-Cas9 mediated amino acids loss.

[1] Department of Epigenetics and Molecular Carcinogenesis, The University of Texas MD Anderson Cancer Center, Smithville, TX 78957, USA. [2] The Center for Cancer Epigenetics, The University of Texas MD Anderson Cancer Center, Houston, TX 77030, USA. [3] Center for Epigenetics, Van Andel Research Institute, Grand Rapids, MI 49503, USA. [4] Department of Bioinformatics and Computational Biology, The University of Texas MD Anderson Cancer Center, Houston, TX 77030, USA. [5] These authors contributed equally: Wei He, Liang Zhang. *email: hxu4@mdanderson.org

Functional screens using CRISPR-Cas9 techniques facilitate the identification of essential genes on a genome-wide scale[1–4]. To fully define the function of protein-coding essential genes, it is necessary to distinguish essential protein domains that directly contribute to cellular phenotypes. In a CRISPR-Cas9 knockout experiment, the sgRNAs that target DNA sequences coding for essential protein domains often result in more significant dropout phenotype compared to other sgRNAs that target the same gene[5]. This is likely because CRISPR-Cas9 introduces small indels that generally lead to either frameshift or in-frame mutations. Frameshift indels tend to abolish protein function, whereas in-frame indels, which result in the gain or loss of amino acids, may or may not impact function depending on where they occur. Proteins with small, in-frame indels in non-essential regions are likely to retain function. In contrast, proteins with indels in essential domains may display compromised protein function due to the disruption of an important structural motif or functional conformation (Fig. 1a). Therefore, a domain-focused CRISPR-Cas9 knockout screen has been proposed to evaluate the functional importance of individual domains, leading the way for in situ protein functional studies[5].

In a pooled high-throughput CRISPR-Cas9 knockout screen, an sgRNA library contains tens of thousands of sgRNAs. This depth of coverage facilitates a tiling-sgRNA design that allows the investigation of domain functions across the entire protein for more than 100 protein-coding genes in a single experiment. Munoz et al. performed the first high-throughput tiling-sgRNA screen on 159 genes and confirmed that the sgRNAs that target pharmaceutically important protein domains are associated with stronger knockout effects[6]. Recently, a method combining tiling-sgRNA screens with positive selections has been developed to identify small-molecule drug target sites[7]. A computational pipeline, CRISPRO, maps functional scores of tiling sgRNAs to genomes, transcripts, protein coordinates and structures, providing general views of structure-function relationships at discrete protein regions[8].

Despite these advances, pooled high-throughput CRISPR-Cas9 screens are subject to inactive sgRNAs, off-target effects, and high noise-to-signal ratios, posing computational challenges to the robust identification of essential domains. To address these challenges, we propose ProTiler, a computational method specifically designed for the analysis of tiling CRISPR screen data. ProTiler significantly reduces the impact of inactive sgRNAs and outlier data points through a two-step denoising procedure and accurately detects regions that are hyper-sensitive to CRISPR knockout using a robust region calling algorithm. Therefore, ProTiler facilitates de novo identification of essential protein domains from CRISPR-Cas9 tiling-sgRNA knockout screens.

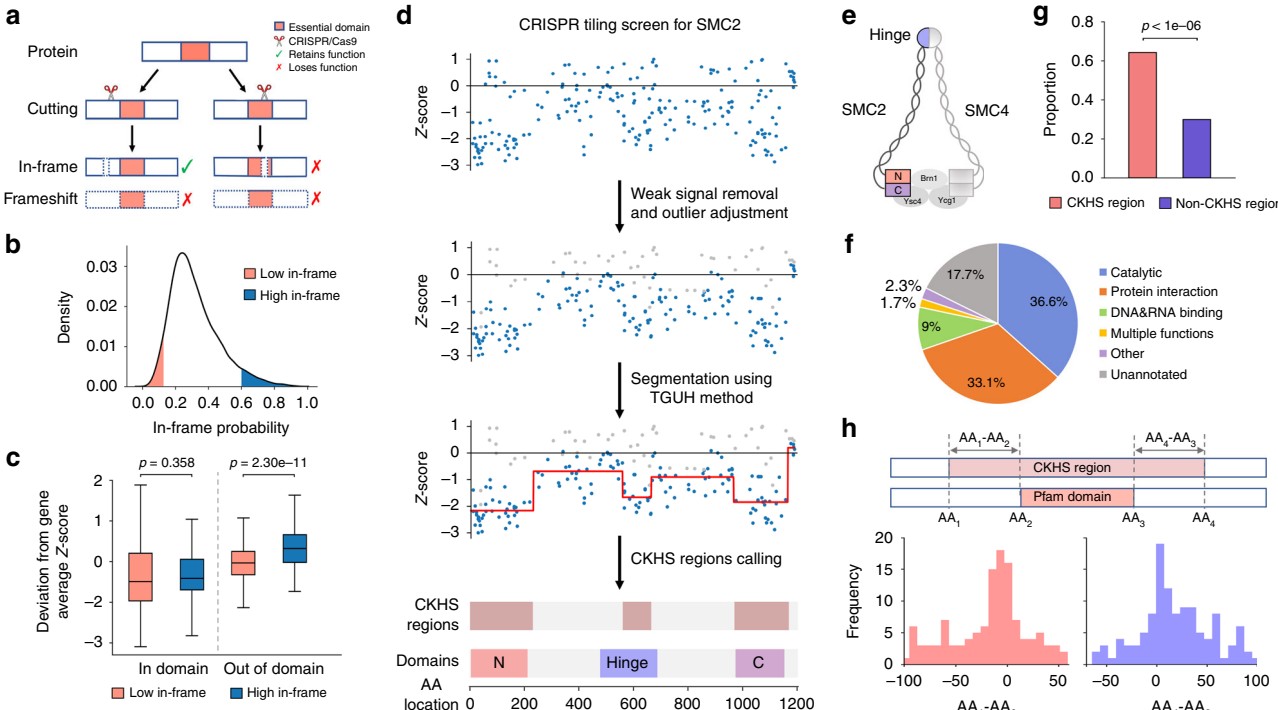

**Fig. 1** Mapping CRISPR-knockout hyper-sensitive (CKHS) regions with ProTiler. **a** An in-frame indel model underlying the rationale of domain-associated CKHS regions. **b** Distribution of in-frame probability of the sgRNAs targeting essential proteins in Munoz data. The probabilities were predicted using inDelphi[11]. The top 5% and bottom 5% sgRNAs are defined as high in-frame (blue) and low in-frame (red), respectively. **c** Box-plots comparing the dropout effects between high in-frame and low in-frame sgRNAs that target proteins containing drug target domains. The center line, bounds of box and whiskers represent the median, interquartile range and 1.5 times interquartile range, respectively. The p-values were computed using the Mann–Whitney test. **d** The workflow of ProTiler. The dot plots show the dropout effects, in Z-score[6] (Y-axis), of sgRNAs targeting the gene coding for SMC2. A negative value of the Z-score corresponds to a dropout effect. Each dot represents an sgRNA mapped to the amino acid location (X-axis). The grey dots and blue dots represent filtered and remaining sgRNAs, respectively. The red line shows the segmented protein regions and their dropout signal levels. **e** A structural model of condensin complex, in which SMC2 and SMC4 form a heterodimer via hinge domains, and their ATPase head domains (N and C) are associated with kleisin subunits to create a ring-like structure. **f** Categorization of CKHS regions based on the molecular functions of overlapped protein domains. **g** A bar chart showing the proportion of AAs in Pfam domains, for CKHS regions and non-CKHS regions respectively. The p-value was empirically computed by random simulation. **h** Distribution of distances between the borders of CKHS regions and domain boundaries as defined in the Pfam database

## Results

**Essential domains are hyper-sensitive to in-frame indels**. The basis of using tiling-sgRNA screens for protein domain analysis relies on an in-frame indel model, as shown in Fig. 1a. Recently, several laboratories showed that CRISPR-Cas9 indel patterns and in-frame mutational probabilities are predictable from sgRNA-targeted DNA sequences[9,10]. Taking advantage of these findings, we first examined if the in-frame indel model (Fig. 1a) is applicable to Munoz data. We used inDelphi[11] to measure the in-frame probabilities for 28,951 sgRNAs corresponding to 108 essential protein-coding genes. The median in-frame probability is 0.29, suggesting incomplete protein knockout in approximately half of the diploid cells. To ensure robustness against prediction error, we selected the sgRNAs that were predicted to be very likely (top 5%, in-frame probability >0.60, high in-frame) or very unlikely (bottom 5%, in-frame probability <0.129, low in-frame) to create in-frame mutations (Fig. 1b). Consistent with the model, the low in-frame sgRNAs showed a greater dropout effect compared to the high in-frame sgRNAs ($p = 4.85\text{e-}76$, Mann–Whitney test, Supplementary Fig. 1). To assess whether the difference between the two categories is associated with the functional essentiality of protein domains, we examined 15 domains targeted by small-molecule compounds that have either been FDA-approved or advanced to clinical trials (Supplementary Data 1). Among the sgRNAs that target DNA sequences coding for these domains, no significant difference was observed between the high in-frame and the low in-frame categories ($p = 0.358$, Mann–Whitney test). In contrast, sgRNAs associated with the regions outside annotated domains showed significant differences ($p = 2.30\text{e-}11$, Mann–Whitney test, Fig. 1c). These results indicate that the essential domains are hyper-sensitive to CRISPR-Cas9 induced in-frame indel mutations, supporting the underlying rationale of using tiling-sgRNA screens to predict domains that are essential.

**ProTiler enables fine-mapping of CKHS regions**. We developed ProTiler, a computational method for the mapping of protein regions that are associated with CRISPR-knockout hyper-sensitivity (CKHS). Figure 1d outlines the major steps in ProTiler, exemplified by tiling-sgRNA screen data for SMC2, a component of the condensin complex. ProTiler first maps sgRNA dropout signals to the amino acids of the target proteins. The data points with weaker dropout effects compared to their neighbors are likely to be associated with inactive sgRNAs (Supplementary Fig. 2), thus are removed. The outliers in the remaining data points can be caused by non-Gaussian variations or additive off-target effects. To reduce the impacts of these outliers, we adjusted their values based on the mean and variation of the surrounding signals. To partition the protein into regions corresponding to different viabilities, we applied Tail-Greedy Unbalanced Haar (TGUH) transformation, a wavelet-based changing point detection algorithm with proven high accuracy and robustness under noisy conditions[11]. Each region was assigned a viability score to be the average of the data points in that region. Finally, we developed an k-mean-like algorithm (k = 2) to classify the regions into CKHS and non-CKHS categories (see Methods section for details). For SMC2, ProTiler detected three CKHS regions, corresponding to the N-terminus, the C-terminus and the middle hinge domain. These CKHS regions are highly consistent with a model of condensin structure, in which SMC2 and SMC4 form a heterodimer via their hinge domains and their ATPase head domains are associated with kleisin subunits to create a ring-like structure[12] (Fig. 1e).

Among 108 essential proteins in the Munoz data, ProTiler-identified 175 CKHS regions in 83 proteins. Those proteins

without an identified CKHS region are associated with relatively smaller size, and often harbor a single domain that covers a majority of the amino acids (Supplementary Fig. 3). In specificity, 82.3% of the 175 regions overlapped with Pfam-annotated protein domains (Fig. 1f, Supplementary Data 3). At the amino acid (AA) level, 64.2% of the AAs in the CKHS regions are within Pfam domains, compared to 30.0% for non-CKHS regions (Fig. 1g). In sensitivity, 207 out of 277 (74.7%) Pfam domains in the 83 proteins were identified. Of note, all of the 15 previously mentioned drug target domains were identified (Supplementary Data 1). Statistically, ProTiler achieved an F1-score of 0.64, significantly higher than random expectation ($p < 1\text{e-}5$, permutation test, Supplementary Fig. 4). To estimate the resolution, we aligned the borders of ProTiler-defined CKHS regions to the boundaries of Pfam domains. The left and right borders of CKHS regions were enriched within 20 AAs from domain boundaries (left: $p = 5.87\text{e-}08$; right: $p = 9.36\text{e-}06$, Fisher exact test) and slightly outside the domains, suggesting that small deletions of AAs adjacent to the domains may also compromise protein function (Fig. 1h). Taken together, these lines of evidence indicate the high performance of ProTiler, in specificity, sensitivity, and resolution.

**CKHS mapping reveals previously unannotated protein domains**. Among the ProTiler-identified CKHS regions, 17.7% did not overlap with any annotated Pfam domains, which may be associated with previously undefined domains. Indeed, some of these regions have been functionally characterized but remain unannotated in the Pfam database. For example, a small CKHS region in the transcriptional coactivator YAP1 (AA86-99) perfectly matches a twisted-coil structure, one of the three independent interfaces that interacts with TEAD1 (Fig. 2a). Consistent with previous findings, our results showed the importance of YAP1-TEAD1 interaction for cell viability, where the region of AA86-99 is the most critical interaction site[13] (Fig. 2b). ProTiler also identified a CKHS region in the N-terminus (AA55-184) of MPS1/TTK, which contains three tetratricopeptide repeat domains that govern localization of the protein to either the kinetochore or the centrosome[14] (Supplementary Fig. 5)

Inspired by these examples, we sought to identify essential domains within newly identified, but unannotated CKHS regions. After careful inspection, we focused on SMARCB1, one of the core subunits of the SWI/SNF chromatin remodeling complex, for further validation. ProTileridentified a CKHS region (AA24-53) in the N-terminus of SMARCB1, in addition to the well-characterized ATP-binding domain. We chose to examine this region because it habors recurrent missense mutations (P48L, R53L, E31V) and in-frame deletions (G29, S30) that have been identified in cancer patients[15] (Fig. 2c). An earlier in vitro study showed that the N-terminus of SMARCB1 forms a putative DNA-binding structure[16], but its function has not been investigated in situ. To validate the function of this region, we constructed vectors expressing either a full-length or an N-terminally-truncated form of SMARCB1 (Fig. 2d, Supplementary Fig. 6a). In the constructs, we introduced synonymous mutations to the ATP-binding domain, such that the knockouts with sgRNAs targeting the mutated sites would affect only the endogenous SMARCB1 without compromising the expression of exogenous proteins. The vectors were lenti-viral introduced into DLD-1, a colon cancer line expressing a wild-type SMARCB1. Consistent with the screen data, CRISPR-mediated knockouts of SMARCB1 hampered the growth of DLD-1 cells (Supplementary. Fig. 6b, c), which could be completely rescued by exogenous expression of the full-length protein, but not the

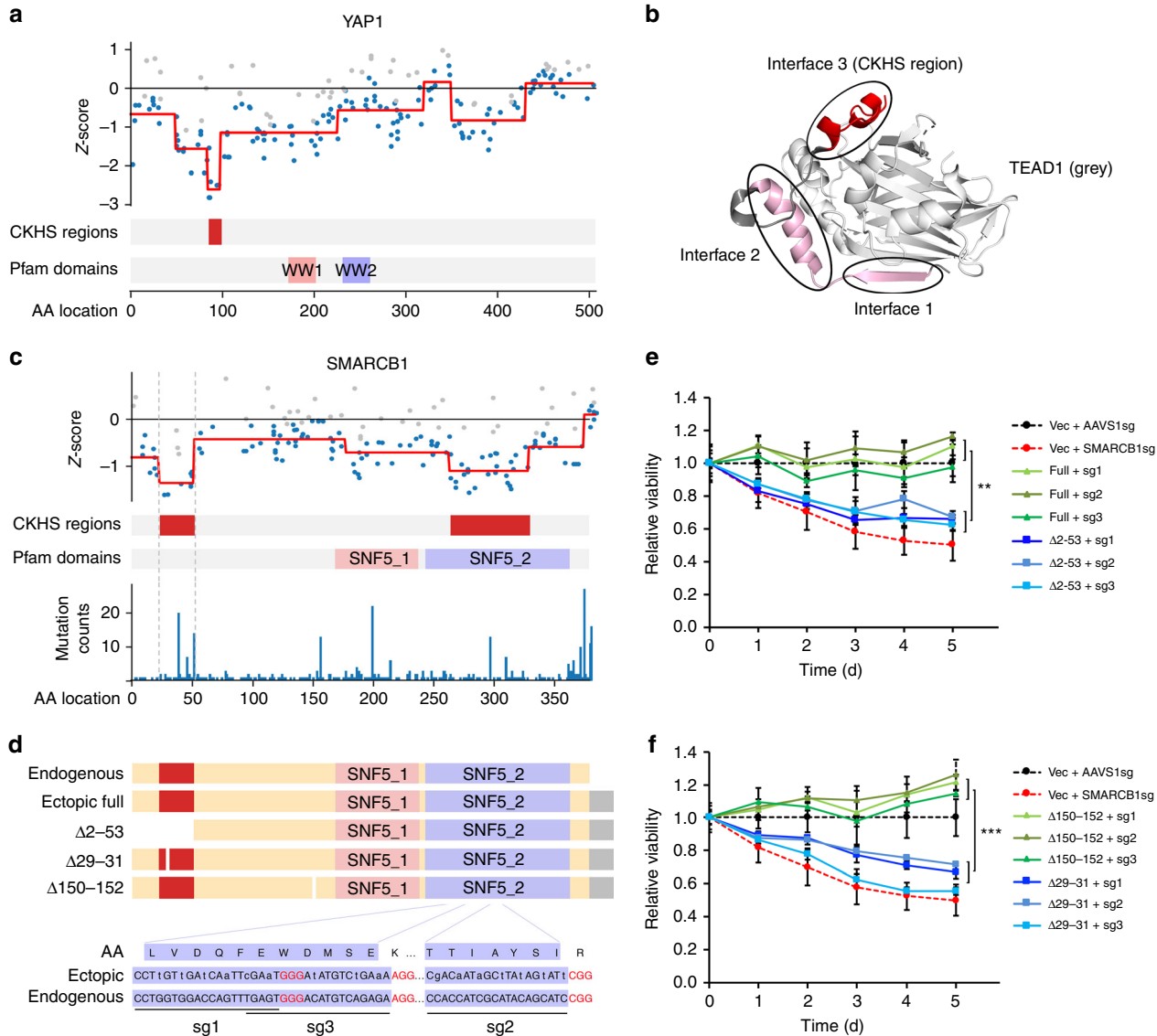

**Fig. 2** CKHS profiling facilitates the identification of unannotated essential domains. **a** The CKHS profile and domain annotation of YAP1. **b** The 3D structure of the YAP1&TEAD1 interaction (PDBID: 3KYS). Three interfaces of YAP1 interact with TEAD1. The CKHS region (interface 3) is highlighted in red. **c** The CKHS profile and domain annotation of SMARCB1, aligned with mutation frequency retrieved from the COSMIC database[47]. **d** A schematic representation of exogenous full-length, a truncated form (Δ2-53), and two small-deletion forms (Δ29-31, Δ150-152) of SMARCB1, as well as the endogenous protein. The CKHS region is highlighted in red. The Myc and 6xHis tag sequences were appended to the C-terminus of exogenous proteins. Three sgRNAs were designed to target the endogenous *SMARCB1* DNA sequence coding part of the SNF5_2 domain. Synonymous mutations were introduced into the exogenous proteins, except the bases at the PAM (red) and methionine codon. **e** Relative proliferation of DLD-1 cells with exogenous expression of full-length or the truncated form of SMARCB1 shown in **d**, in combination of endogenous SMARCB1 knockout. **f** Relative proliferation of DLD-1 cells with exogenous expression of SMARCB1 harboring small deletions inside or outside the CKHS region (Δ29-31 or Δ150-152), in combination of endogenous SMARCB1 knockout. The red and black dash lines represent normalized relative viability of vector control cells with AAVS1sg or SMARCB1sg, respectively. All the data points represent the relative viability normalized to the AAVS1 control group. The error bars represent the standard divation of three biological replicates performed at each time point. The star symbols represent statistical significance: $p < 0.01$ (**) and $p < 0.001$ (***). The p-values were computed using *t*-test. Source data are provided as a Source Data file

truncated protein (Fig. 2e). To further test if a small deletion in the CKHS region is sufficient to abolish SMARCB1 function, we expressed the SMARCB1 protein harboring a 3AA deletion in the CKHS region (Δ29-31), as well as a control where a deletion of the same size was introduced to the non-CKHS region (Δ150-152) (Fig. 2d, Supplementary Fig. 6a). As expected, the Δ150-152 protein, but not the Δ29-31 protein, rescued the phenotype of endogenous SMARCB1 knockouts (Fig. 2f). Collectively, these lines of evidence confirmed the essential role of the CKHS region in the N-terminus of SMARCB1.

**CKHS profiling of multi-domain proteins**. Proteins containing multiple functional domains often have complex cellular roles, thereby posing challenges to their molecular characterization. A tiling-sgRNA screen can facilitate the assessment of domain functions in a high-throughput manner. We explored the CKHS profiles of 51 multi-domain proteins in the Munoz dataset. Domains in these proteins are associated with a wide range of functions, including catalysis, protein–protein interactions, and DNA/RNA binding. For these 51 proteins, 62.1% of the domains were marked with CKHS regions (Supplementary Data 4).

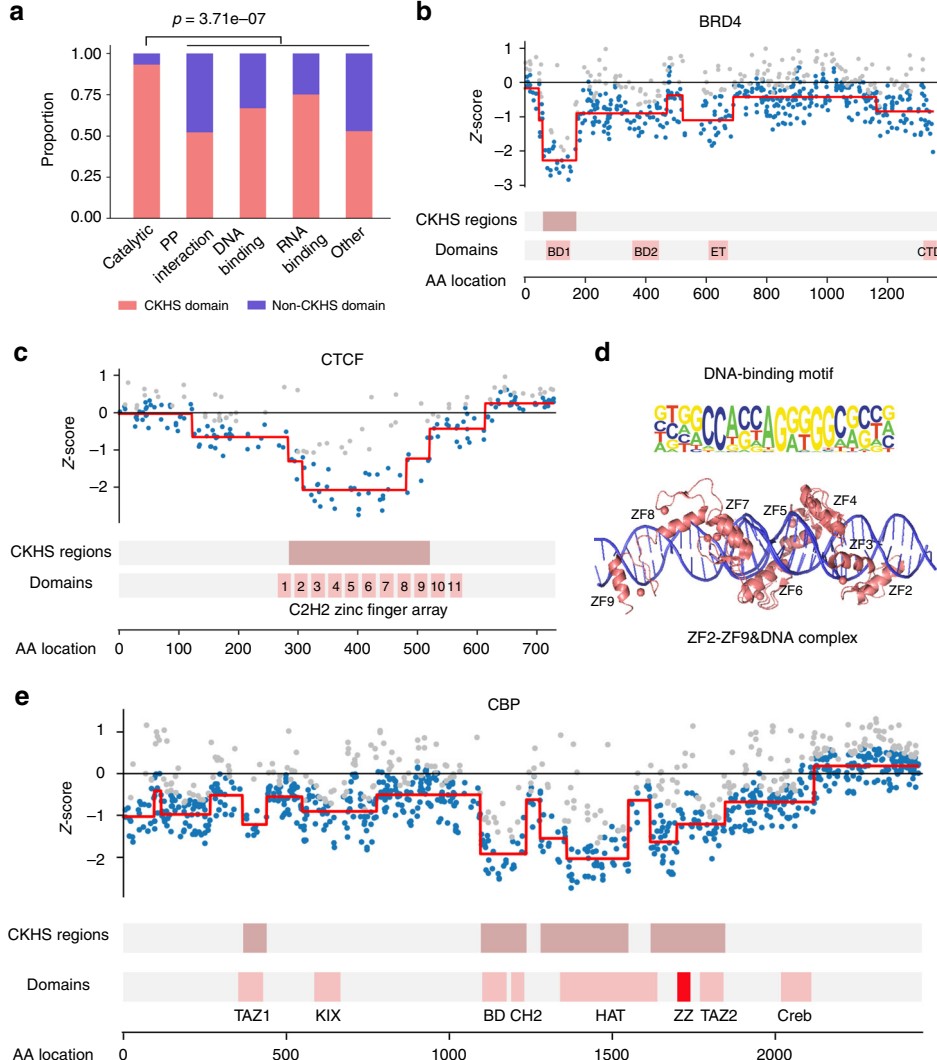

**Fig. 3** The CKHS profiles of multi-domain proteins. **a** All the domains in 51 multi-domain proteins were categorized based on their molecular functions. The bar chart shows the proportion of CKHS-overlapped domains for each category. The p-value was computed based on hypergeometric distribution. **b** The CKHS profile and domain annotation of BRD4. **c** The CKHS profile and domain annotation of CTCF. **d** The 3D structure of CTCF (ZF2-ZF9) complexed with DNA (PDB IDs: 5UND;5T0U). The DNA-binding motif[48] is aligned with the structure. **e** The CKHS profile and domain annotation of CBP. The ZZ domain is highlighted in red

Catalytic domains had the highest likelihood (93.2%) of being essential, compared to the others (55.0%, $p = 3.71e-07$, Fisher exact test, Fig. 3a). Although it is beyond the scope of this study to elucidate the mechanism underlying the domain essentiality for each protein, we sought to link CKHS profiles to recent findings in the literature, as a proof of value for novel protein functional discovery. Here, we present three examples.

BRD4 is an acetyl-lysine reader known for its function as a transcriptional coactivator[17]. It is also a major target of several BET inhibitors that have progressed to clinical trials[18]. BRD4 contains two bromodomains (BD1 and BD2), an extra-terminal domain (ET), and a C-terminal domain (CTD). Despite high sequence similarity between BD1 and BD2, only BD1 showed hyper-sensitivity to CRISPR knockouts (Fig. 3b). In line with this observation, an earlier investigation reported that inhibition of BD2 had a milder effect on BRD4-dependent gene transcription than BD1 inhibition[19]. Recently, several laboratories reported a BD1-dependent role of BRD4 in the DNA damage response pathway, raising additional concerns regarding BRD4 function involved in the viabilities of normal and cancer cells[20–22]. To this end, our results highlighted the importance of bromodomain

selectivity in the functional analysis and drug discovery regarding BRD4 inhibition.

CTCF is a DNA-binding protein critical for maintaining high-order chromatin conformations[23]. It has 11 adjacent zinc-finger domains (ZF1-ZF11) that bind to DNA. Its CKHS profile showed that only ZF2-ZF9 are hyper-sensitive, suggesting that individual ZFs have unequal contributions to CTCF function (Fig. 3c). Recently, a crystal structure of CTCF-DNA interaction was released, which showed that ZF2-ZF9 are required for the binding to the full nucleotide motif sequence[24] (Fig. 3d). Thus, our high-resolution CKHS map is supported by this structure model, and in turn reinforces the new structural model in situ.

CBP, a paralog of p300, is a transcriptional coactivator that has been extensively studied over the last two decades[25]. It has a catalytic core containing a HAT domain, a bromodomain, and a CH2 region[26]. All the three regions were found to be CKHS in our results (Fig. 3e). Two protein-interacting domains, TAZ1 and TAZ2, were also sensitive to CRISPR knockouts but to a lesser degree. In addition to these well-characterized domains, a ZZ-type zinc-finger domain adjacent to the HAT domain is also in the CKHS region. Recently, the ZZ domain was characterized to

be an acetyl-reader of histone H3, which modulates CBP/p300 enzymatic activity and their associations with chromatin[27]. Therefore, the CKHS profile of CBP further supports the critical role of the ZZ domain.

Taken together, these examples indicate that the CKHS profiles can be used either to infer a new functional model, or to validate an existing model. Potential applications include, but are not limited to, prediction of potent inhibitor targets, discovery of alternative protein functions, in situ validation of protein structure, and identification of novel domain function.

**CKHS regions are correlated with PTMs**. Despite the consistency between CKHS regions and essential domains in CBP, we observed that a small region in the HAT domain (AA1550-1618) was insensitive to CRISPR knockouts. This region matches the auto-acetylation sites that are known to regulate the catalytic activity of CBP[28] (Fig. 4a). Similarly, a cluster of phosphorylation sites between the kinase domain and the RAS-binding domain of BRAF also showed hypo-sensitivity despite their critical roles in regulating BRAF activity[29]. Another group of phosphorylation sites near AA600 of BRAF is in the CKHS region, but showed lesser sensitivity compared to the adjacent kinase domain (Fig. 4b). Indeed, when we mapped the post-translational modification (PTM) sites onto our data, we found phosphorylation and acetylation sites were significantly depleted inside CKHS regions compared to outside CKHS regions (Fig. 4c). Therefore, the sensitivity to CRISPR knockouts is negatively correlated with phosphorylation and acetylation sites. Since those PTMs are often clustered to modulate protein conformation via electric charges, a possible explanation to this observation is that a CRISPR-Cas9 mediated small deletion of amino acids near PTMs does not significantly reduce charges in a local region, which in turn has weaker impact on protein conformation and phenotype. Additionally, methylation and ubiquitination are also statistically associated with CKHS regions, but the explanations for these associations are yet elusive. Collectively, these observations indicate that protein domains and PTM sites have distinct sensitivities to CRISPR-Cas9 mediated loss of amino acids; therefore, a careful inspection of PTM annotations will be helpful for the interpretation of tiling-sgRNA screen data.

**CKHS regions are predictable from protein features**. The sgRNA knockout effects are associated with both the level of amino acid sequence conservation and the secondary structure of target protein regions[8]. CRISPRO introduced a machine-learning approach for the prediction of CRISPR-Cas9 knockout effects, using the features related to protein function and sequence-specific sgRNA activity. Consistently, we found that CKHS regions are associated with highly conserved regions (SIFT score) and molecular secondary structures (Fig. 4d, e). Since the tiling screen data are available for only a limited number of proteins, we sought to predict CKHS regions based on protein features in a proteome-wide scale. Recently, several variants of the CRISPR system have been developed for genome editing[30–32], each associated with a unique sequence preference for sgRNA activity. Therefore, we aimed at a predictive model based on protein features alone, such that the predicted CKHS regions would be independent of the sgRNA target sequences and could be used for different CRISPR techniques.

Using a bagging Support Vector Machine (bagging SVM)[33] and a leave-one-gene-out cross-validation strategy, we first examined the predictive power of individual features using the Munoz data. The SIFT score showed the highest power (AUC = 0.713), followed by domain (AUC = 0.650) and PTM (AUC = 0.632) annotations (Fig. 4f). As references, transcript variant

coverage and AA position in relative to protein N- and C-termini have little predictive power, indicating that protein functional features are the determinants of CKHS profiles. Our model further achieved a higher performance when it integrated SIFT score, secondary structure, protein domain, and PTM annotations (Fig. 4g). Notably, excluding domain annotations did not significantly compromise the prediction, suggesting that this approach can be applied to the proteins lacking domain information. A proteome-wide prediction of CKHS regions is available in Supplementary Data 5.

To check if the SVM model can predict the outcome of other tiling screen data, we compared the SVM predictions to the CKHS regions identified from an independent tiling screen dataset that includes two genes: ZBTB7A and MYB[8]. We observed high consistency among the SVM predictions, ProTiler-identified CKHS regions, and the Pfam-annotated protein domains (Supplementary Fig. 7), which implies the predictive power of our model beyond the scope of Munoz data. We further compared the SVM-predicted regions to two recent reports of new functional domains: a non-catalytic FLOS domain in methyltransferase SETD1A and an eMIC domain in splicing factor SRRM4[34,35]. In both proteins, the SVM model correctly predicted the novel domains (Supplementary Fig. 8). Taken together, these results suggest that the proteome-wide CKHS region prediction provides insights into novel domain discovery when tiling screen data are unavailable. Meanwhile, we note that the SVM prediction is imperfect (AUC between 0.7–0.8); therefore, additional proteomic data and functional validations are still needed for accurate domain discovery.

Finally, we tested whether the predicted CKHS regions can improve the sgRNA design, using two large-scale CRISPR-Cas9 screen datasets based on the GeCKO-v2 and Avana libraries in which 4-6 sgRNAs were designed for each gene[36,37]. Our results showed that the selection of sgRNAs based on the CKHS predictions achieved greater dropout effects for known essential genes, suggesting the potential of applying our model to the rational design of CRISPR sgRNA libraries (Fig. 4h and Supplementary Fig. 9).

## Discussion

Essential protein domains are often associated with hyper-sensitivity to CRISPR-Cas9 knockouts, permitting in situ analysis of protein functions. Previous applications have mainly focused on the validation of annotated or hypothetical domains[5,7]. A pooled CRISPR screen allows a tiling-sgRNA design for more than 100 proteins, holding promise for de novo discovery of essential domains. Here, we propose ProTiler for the mapping of CKHS regions from a tiling-sgRNA CRISPR screen. The high performance of ProTiler enables the identification of multiple essential domains in a protein, exemplified by SMC2, CTCF, and CBP in this article. Using ProTiler, we identified an unannotated essential domain in the N-terminus of SMARCB1, which was further experimentally validated. The CKHS regions are highly concordant with essential domains of different types, including those associated with catalysis, protein–protein interactions, and DNA or RNA binding. These results established the tiling-sgRNA approach as a general method for in situ protein functional studies. The tiling-sgRNA screens applied on panels of uncharacterized proteins will greatly expand our knowledge on novel protein and domain functions in a proteome-wide scale. Of note, although this work is focused on identifying essential domains that contribute to cell viability, our approach can also be applied to the screens with other readouts or cell sorting systems to understand the domain functions associated with various cellular phenotypes.

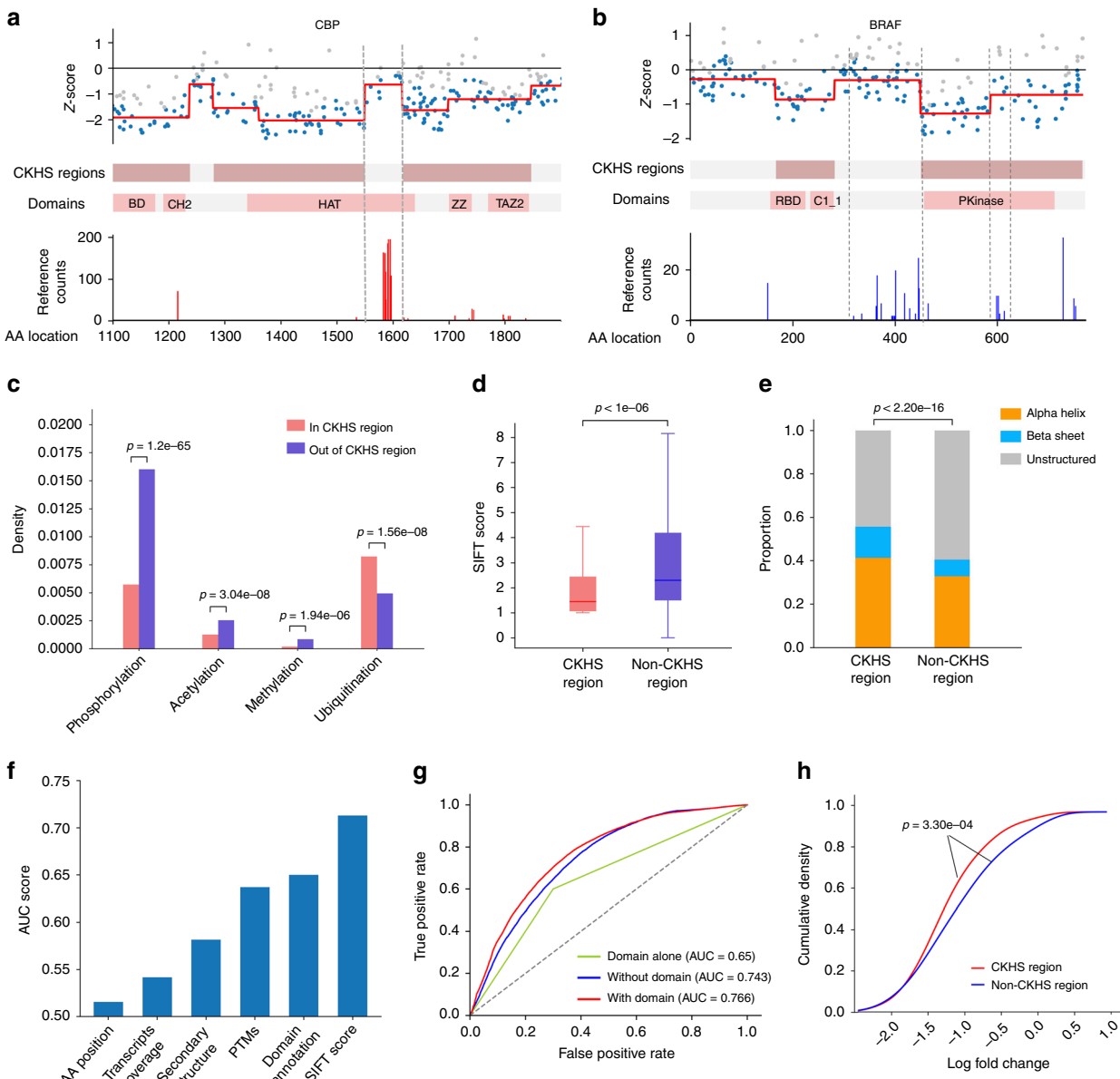

**Fig. 4** CKHS regions are are predictable from protein features. **a** The zoomed-in views of CKHS profile, domain annotation, and acetylation sites of CBP (AA1100-1900). The Y-axis of the PTM profiles represents the number of publication references collected at https://www.phosphosite.org. **b** The CKHS profile, domain annotation, and phosphorylation sites of BRAF. **c** A bar chart showing the density of PTM sites inside or outside of the CKHS regions. The p-values were computed based on hypergeometric distribution. **d** A box-plot showing the association between CKHS regions and amino acids conservation (SIFT score). The center line, bounds of box and whiskers represent the median, interquartile range and 1.5 times interquartile range, respectively. The p-value was computed using the Mann–Whitney test. **e** A bar chart showing the distribution of secondary structures for CKHS and non-CKHS regions. The p-value was computed using the Chi-square test. **f** A bar chart showing the predictive power of bagging SVM, in ROC-AUC score, using individual protein features. The AA position in the protein and the transcript coverage are used as references. **g** ROC curves showing the predictive powers using all protein features, all features other than domain annotation, and domain annotation alone, respectively. **h** The sgRNAs targeting core essential genes were categorized based on predicted CKHS regions. The cumulative distributions of sgRNA dropout effects in Avana dataset[37] are shown for each category. The p-value was computed using Kolmogrov–Smirnov test

CRISPRO is the first computational tool for mapping the sgRNA functional scores to proteins[8]. Compared to CRISPRO, ProTiler includes an algorithm to reduce the impacts of sgRNA activity and outlier data points, minimizing the occurance of discontinued segments, and false-positives (Supplementary Fig. 10a). In addition, ProTiler has a robust region calling algorithm to facilitate a de novo analysis. A quantitative comparison of these two methods on the Munoz data showed that ProTiler outperforms CRISPRO in sensitivity, specificity, and resolution (Supplementary Fig. 10b). On the other hand, CRISPRO is advantageous in providing an integrative view of genomic, transcriptomic, proteomic, and structural information, and it is suitable for the analysis of well-characterized proteins.

We showed that the observation of CKHS regions is a consequence of in-frame indel mutations that influce protein domain function (Fig. 1c). It is possible that some protein-independent factors, such as transcription variants, reverse transcription, enhancers, and RNA modification sites, also contribute to the knockout hyper-sensitivity. In our study, neither the variant coverage nor AA position had sufficient power to predict the

CKHS regions, suggesting protein domain function is the major determinant of CKHS regions (Fig. 4f). Meanwhile, we observed a weak but statistically significant association between the coverage of variants and the CKHS regions (Supplementary Fig. 11). Therefore, it would be helpful to check other possible confounding factors when interpreting data arising from a tiling-sgRNA screen.

A typical design of a CRISPR tiling-sgRNA library includes all the target sequences followed by a PAM motif. Screens with such an all-target library are subject to inactive sgRNAs and off-target effects. ProTiler addresses these computational challenges by removing weak signals and adjusting outliers. We note that a number of computational tools have been developed to predict on-target activities and off-targets[38–40]. These tools are useful for designing sgRNA libraries for genome-scale screens. However, filtering sgRNAs according to computational predictions will compromise the overall resolution of a tiling-sgRNA library due to a considerable number of unavoidable computational false negatives. Similarly, sgRNA selection based on the prediction of in-frame probabilities may not be applicable for the design of tiling-sgRNA libraries because only 11.2% of sgRNAs have a predicted in-frame probability greater than 0.5 (Fig. 1b). Therefore, an all-target or nearly-all-target library seems to be more plausible, where the maximum information is conserved. At the same time, it is likely that ProTiler can be further improved using an algorithm that prioritizes sgRNAs based on such computational predictions.

## Methods

**Datasets and external software.** The Munoz dataset was retrieved from publication[6]. The Pfam domain annotation was downloaded from the Pfam database[41] (version 31.0). The PTM annotation was downloaded from the PhosphoSitePlus database[42]. The Avana dataset and the GeCKO dataset were downloaded from https://figshare.com/articles/DepMap_Achilles_19Q1_Public/7655150 and https://figshare.com/articles/DepMap_GeCKO_19Q1/7668407, respectively. The transcript annotation was downloaded from the NCBI CCDS database[43]. The list of 15 drug targets were manually curated in reference to the publications in Supplementary Data 1. The in-frame probabilities were computed using the inDelphi online version at https://indelphi.giffordlab.mit.edu/. The amino acid conservation scores were computed using SIFT[44]. The protein secondary structures were computed using RaptorX[45].

**Identification of essential genes.** The Munoz dataset includes tiling-sgRNA screens on three cell lines (RKO, NCI-H1299, and DLD-1). We computed the average Z-score for each gene in each cell line. Using a threshold of -0.4, as suggested in the original publication, we identified 80, 87, and 90 essential genes for RKO, NCI-H1299, and DLD-1 cell lines, respectively (Supplementary Data 2). The union of the three contains 108 essential genes. If a gene is essential in more than one cell line, we averaged the Z-scores for each sgRNA to increase the signal-noise ratio.

**The ProTiler algorithm.** ProTiler takes three steps to identify CKHS regions: (1) weak signal removal and outlier adjustment; (2) segmentation using TGUH method; (3) CKHS region calling from segments.

Approximately 1/3 of the sgRNAs with a PAM-appended target are inactive, corresponding to weak dropout effects[38,46]. To remove weak dropout signals, each sgRNA data point is compared to its $k$ neighbors to the left and $k$ neighbors to the right. The data point is removed if the signal is weaker than 2/3 of left neighbors and 2/3 of right neighbors. We set $k = 5$, corresponding to an average window span of ~30 AAs, the size of the smallest protein domain module. To adjust the outliers, we estimated the variation of noise for each protein, by applying Median Absolute Deviation (MAD) on the differences between consecutive sgRNA signals. For each data point $x$, we compare it to the median value of its neighbors within a sliding window of size 11. If $x$ is larger than the median value by more than twice of MAD, $x$ is marked to be an outlier and is adjusted to be median + 2*MAD. The outliers below the median values are detected and adjusted in a similar way.

To segment the protein into regions corresponding to hyper- or hypo-sensitivity, ProTiler uses a Tail-Greedy Unbalanced Haar (TGUH) method, which decomposes noisy 1-D data and detects multiple change-points based on wavelet transformation[11]. Different from regular binary segmentation methods that adopt a top-down strategy to search for segments, TGUH uses a bottom-up strategy via a natural unary-binary tree, making it more accurate in recognizing small segments. In ProTiler, TGUH is implemented using R-package library "breakfast".

We used a k-mean-like algorithm (k = 2) for CKHS region calling. Suppose we have $n$ segments, and the $i$th segment contains $k_i$ data points ($i = 1, 2, …, n$). We assigned a score, $s_i$, to be the average of data points in the $i$th segment. The segments were sorted in ascending order of scores, such that $s_1 \leq s_2 \leq … s_n$. Since a more negative value corresponds to a stronger dropout effect, the sorted list is in descending order of CRISPR-knockout sensitivity. We iteratively assign the segments to the CKHS category. The pseudo-code of CKHS region calling is as follow:

```
BEGIN
    assign the first segment to be CKHS
    for i in 2 to n
        compute m_assigned = 1/(i-1) Σ_{j=1}^{i-1} s_j k_j
        compute m_unassigned = 1/(n-i) Σ_{j=i+1}^{n} s_j k_j
        if s_i < (m_assigned + m_unassigned)/2
            assign the ith segment to be CKHS
        else
            break
    merge adjacent CKHS segments in the protein into a single
    CKHS region
END
```

**Essential region calling using CRISPRO.** CRISPRO is a computational pipeline that maps functional scores associated with guide RNAs to genomes, transcripts, and protein coordinates and structure for the analysis and visualization of tiling CRISPR screen data. Although the method was not specifically designed to call protein domains, it utilized LOESS regression to smooth the tiling screen signal and thus helps prioritize functionally important protein regions. To compare ProTiler and CRISPRO in detecting essential protein domains, we applied CRISPRO to the Munoz dataset to call essential protein regions. Specifically, for each essential gene in the dataset, we defined the essential amino acids to be those with CRISPRO LOESS regression scores below a threshold. For a fair comparison, the threshold was determined to allow the same number of essential residues called by both CRISPRO and ProTiler. The essential residues called using CRISPRO were then merged into regions if their distance was less than 3AAs.

**Metrics used for the comparison between CRISPRO and ProTiler.**

- **# Indentified AAs:** The number of amino acids within the identified CKHS regions.
- **Precision:** The proportion of amino acids in the identified CKHS regions that are located within annotated Pfam domains.
- **Recall:** The proportion of amino acids in the annotated Pfam domains that are located within identified CKHS regions.
- **F1-score:** The F1-score is calculated using the following formula:

$$F1 = 2 * \frac{Precision * Recall}{Precision + Recall}$$

- **# Identified regions:** The total number of CKHS regions called.
- **% Overlaps with Pfam domains:** The percentage of CKHS regions that overlap with annotated Pfam domains.
- **% Pfam domains identified:** The percentage of annotated Pfam domains that overlap with identified CKHS regions.
- **% Regions within 20 AAs from left borders of Pfam domains:** The percentage of identified CKHS regions with left borders within 20 AAs of the left boundaries of annotated Pfam domains.
- **% Regions within 20 AAs from right borders of Pfam domains:** The percentage of identified CKHS regions with right borders within 20 AAs of the right boundaries of annotated Pfam domains.

**Prediction of CKHS region.** The protein features were extracted and encoded as follows:

- **Domain annotation:** Each AA is assigned 1 if it is within a Pfam domain, otherwise assigned 0.
- **Conservation score:** SIFT score was computed for each amino acid, followed by a Gaussian kernel smoothing with bandwidth = 10AA. The conservation scores were mean-centered for each protein.
- **Secondary structure:** The secondary structures were predicted using RaptorX and were mapped to each AA. We assigned a code of [1,0] for alpha helix, [0,1] for beta sheet, and [0,0] for unstructured.
- **PTM:** The annotations of phosphorylation, acetylation, methylation, and ubiquitination were mapped to the protein, followed by a Gaussian kernel smoothing with bandwidth = 10AA.

A bagging SVM model was implemented for prediction[33]. The model contains 100 SVMs. In each iteration of bootstrapping, 5% of the AAs were randomly selected with replacement from the CKHS regions, and the same number of AAs were selected from the non-CKHS regions. The final prediction score is the average

of the outputs from all SVMs. The SVMs were implemented using R-package library "e1071".

To predict CKHS region in a proteome-wide scale, the bagging SVM model was trained using the CKHS regions identified in the Munoz dataset, and was applied to all the CDS proteins. Predicted CKHS regions were further merged if their distance is less than 3 AAs. To reduce false positives, regions shorter than 10 AAs were discarded. In the analysis of the Avana and GeCKO datasets, the sgRNA dropout effects were estimated using CERES[37], and were averaged across the cell lines. Two hundred twelve predefined core essential genes[36] were used For Fig. 4h and Supplementary Fig. 5.

**Cell culture.** The colon cancer cell line DLD-1 was obtained from ATCC (CCL-221). Cells were maintained in RPMI-1640 medium with 10% fetal bovine serum (FBS) and 1% Penicillin-Streptomycin. Medium was refreshed every 2–3 days. HEK293T cells were cultured with DMEM medium supplemented with 10% FBS and 1% Penicillin-Streptomycin.

**sgRNA design and nucleotide modification of the ectopic SMARCB1.** The tiling sgRNA sequences of SMARCB1 were obtained from the previous publication[6]. The top three sgRNAs targeting the SNF5_2 domain with most negative z-scores were chosen for further experiments. To ectopically express of SMARCB1, the third bases of codons except the bases at PAM sites and methionine codon in the sgRNA targeting sequences were switched to other bases without coded amino acids changes. The tandem sequences coding Myc and 6xHis tags were appended at the 3' terminus of the modified SMARCB1. To generate truncated SMARCB1, the nucleotides coding position 2–53 amino acids were removed on the basis of mutant full-length SMARCB1.

**Plasmid construction.** Human mutant full-length and truncated SMARCB1 were synthesized with modified nucleotides (Biomatik, USA). To construct the over-expression plasmid, the mutant full-length and truncated fragments with Myc and His tags were, respectively, subcloned into a pLVX-IRES-tdTomato vector (#631238, Clontech, USA) using restriction sites XbaI and BamHI. SMARCB1 mutants with small AA deletions (ΔG29,S30,E31 and ΔD150,K151,K152) were produced by site-directed mutagenesis on the basis of mutant full-length SMARCB1 (#E0554S, NEB). Each mutant was verified by sequencing, and protein expression was confirmed by Western blot. To knockout endogenous SMARCB1, sgRNA oligos were synthesized (Sigma, USA) and cloned into lentiCRISPRv2 (#52961, Addgene) according to the protocol from Feng Zhang's lab. The sgRNAs targeting AAVS1 gene were used as the controls.

**Virus packaging and infection.** HEK293T cells ($4 \times 10^6$) were seeded into 10 cm cell culture dishes 1 day before transfection in fresh medium. Before transfection, 4 μg target plasmid, 4 μg psPAX2 and 2 μg pMD2.G plasmids were added in 1 mL pre-warmed Opti-MEM medium (#31985062, Gibco), and then mixed with 24 μL X-tremeGene HP DNA Transfection Reagent (#6366236001, Roche) at room temperature for 30 min. The mixture was dropwise added into each 10 cm dish containing HEK293T cells. Virus supernatant was collected 48 h after transfection, filtered through a 0.45 μm Acrodisc syringe filter, frozen in small volume and stored at −80 °C until use. For infection, cells were seeded into six-well plates with $5 \times 10^5$ cells/well. After cells attached, lentivirus and 2 μL polybrene (#TR-1003-G, Millipore) were added with totally 2 mL medium in each well. Forty-eight hour after infection, cells were seeded into 10 cm dishes for Puromycin (2 μg/mL) selection. To determine multiplicity of infection (MOI), different volumes of lentivirus were used for infection. Cell survival rate was calculated after Puromycin selection.

**Knockout and ectopic expression of SMARCB1.** To test the knockout effects of sgRNAs targeting SMARCB1, viral lentiCRISPRv2-AAVS1sg and lentiCRISPRv2-SMARCB1sg were used for infection, followed by 3–7 days of Puromycin selection. To ectopically express mutant SMARCB1, DLD-1 cells were infected with lentivirus harboring pLVX-IRES-tdTomato, pLVX-IRES-tdTomato-mutFullSMARCB1, pLVX-IRES-tdTomato-mutTruncSMARCB1, and pLVX-IRES-tdTomato-mutΔS-MARCB1, cultured for several days and then sorted by FACS, respectively. For the knockout rescue experiment, the sorted cells were infected with viral lentiCRISPRv2-AAVS1sg or lentiCRISPRv2-SMARCB1sg for SMARCB1 knockout and subsequently selected by Puromycin.

**Western blot.** The knockout and overexpression of SMARCB1 were verified by western blot. The cells sorted by FACS and 1 week after lentiCRISPRv2-SMARCB1sg virus infection were collected for protein extraction. The cells with pLVX-IRES-tdTomato and lentiCRISPRv2-AAVS1sg were used as the controls. Proteins were separated by SDS-PAGE and transferred onto PVDF membranes by a semi-dry transferring system (Bio-rad, USA). After blocking with 5% skimmed milk, membranes were incubated with primary antibodies of rabbit anti-SMARCB1 (#A301-087A-M, Bethyl), anti-His (#12698 T, CST), and anti-Myc (#2278 T, CST) at the concentration of 1:2000 at 4 °C overnight, respectively. After TBST washing, secondary anti-rabbit (#NA934, GE) and -mouse (#NA931, GE) antibodies were

used for incubation at 1:10000 for 1–2 h. The bands were visualized by ECL reagent (#NEL104001EA, PerkinElmer). β-actin was used as the loading control.

**Cell growth assay.** After 3–7 days of Puromycin selection, the cells infected with lentiCRISPRv2-SMARCB1sg virus were seeded into 96-well plates at the density of 1000 cells/well in 100 μL medium. Cell proliferation was analyzed based on viability as determined by CellTiter-Glo (#G7572, Promega) according to the manufacturer's instructions. Triple independent repeats were conducted.

**Reporting summary.** Further information on research design is available in the Nature Research Reporting Summary linked to this article.

## Data availability
The processed data are available in the Supplementary Data of this article. The graphic views of CKHS profiles of 83 proteins in the Munoz dataset are publicly available at https://figshare.com/s/27d52df30b2bcc7a038a. The source data underlying Fig. 2e, f and Supplementary Fig. 6a–c are provided as a Source Data file. All other relevant data can be obtained from the authors upon reasonable request.

## Code availability
ProTiler was written in Python (version 2.7) and R package (version > 3.5.0), implemented as open source software downloadable from https://github.com/MDheweir/ProTiler-1.0.0.

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

## Acknowledgements

We thank Drs. Sharon Dent and Xiaodong Cheng for critical discussion, and Dr. Briana Dennehey for manuscript inspection. This work was supported by a CPRIT grant RR160097 (H.X.). H.X. is a CPRIT Scholar in Cancer Research.

## Author contributions

W.H., L.Z. and H.X. conceptualize the study. W.H. developed the software. W.H., O.D.V. and J.D. performed computational analysis. L.Z., R.F., E.B., and A.Y.P. performed bio-chemistry experiments. M.T.B, X.S., T.C. and B.B. helped data interpretation. H.X. supervised the project. All authors participate in writing the manuscript.

## Competing interests

The authors declare no competing interests.
