## [Peer Review File · Nature Communications]

Reviewers' Comments:

Reviewer #1:

Remarks to the Author:

In this paper He et al. present ProTiler, a computational method to analyze tiling data from CRISPR/Cas9 knockout screen aimed at uncovering protein domains.

The manuscript is well written and the software well documented. Interestingly the method identified unannotated domains of SMARCB1 that were subsequently validated by the authors.

I think this manuscript needs only minor revisions and it could benefit addressing the following points before publication:

1) Line 43: "This is likely because CRISPR/Cas9 introduces small indels that create frameshift or in-frame mutations in a stochastic manner." This contradicts the recent prediction models that can predict alleles after DSBs, one of which is used in this paper (inDelphi, lines 77-79).

2) Despite a competing method called CRISPRO is mentioned in the introduction no comparison with ProTiler is presented. I think it is important to directly and quantitatively compare the two methods with the Munoz dataset and highlight unique and common features.

3) "To ensure robustness against prediction error, we selected the sgRNAs that were predicted to be very likely (top 5%, in-frame probability > 0.60, "high in-frame") or very unlikely (bottom 5%, in-frame probability < 0.129, "low in-frame") to create in-frame mutations (Fig. 1b)."

I think it would be really helpful to know how this affects the tiling resolution. Could you please create a plot showing the distribution of the distances of consecutive sgRNAs before and after this filtering?

4) "The data points with weaker dropout effects compared to their neighbors are likely to be associated with inactive sgRNAs (Supplementary Fig. 2), thus are removed." Briefly explain how this filter is implemented/what is the threshold.

5) "We adjusted their values based on the mean and variation of the surrounding signals." See the previous point. Is this a smoothing?

6) "Finally, an iterative algorithm classifies the regions into CKHS and non-CKHS categories." See 4) and 5). Add a very short sentence to explain what is the key idea of this algorithm.

7) "Among 108 essential proteins in the Munoz data, ProTiler identified 175 CKHS regions in 83 proteins. 82.3% of these regions overlapped with Pfam annotated protein domains" and "At the amino acid (AA) level, 64.2% of the AAs in the CKHS regions are within Pfam domains, compared to 30.0% for non-CKHS regions". Please report also the opposite i.e. how many annotated Pfam protein domains in the analyzed genes were missed by ProTiler.

8) "The borders CKHS regions were enriched within 20 AAs from domain boundaries and slightly outside the domains". Please provide a quantitative measure for the enrichment. Also, a simple measure to assess performance would be the F1-score. In fact, this problem can be thought as a binary classification problem: each aminoacid in the protein is a domain (1) or not (0).

Reviewer #2:

Remarks to the Author:

In this study He et al. describe how a tiling sgRNA library can be applied to identify essential

protein domains. They developed an algorithm named ProTiler to map hyper-sensitive CRISPR knockout regions based on the severity of sgRNA dropout from a tiling screen. They applied ProTiler on a published tiling screen dataset from Munoz et al. and identified hypersensitive regions that largely overlap with annotated Pfam domains as well as known drug target domains in proteins from this dataset. They also identified a new domain in SMARCB1 and predicted hypersensitive regions in a proteome-wide scale based on protein features.

Previously Schoonenberg et al. developed a similar algorithm named CRISPRO for which the focus appears more on the visualisation of essential domains in the protein structure and on prediction for the efficiency of guides. However, Schoonenberg et al. also describe the capacity of CRISPRO to highlight important protein regions and help prioritize protein regions of interest for chemical biology. Although CRISPRO also enables to detect essential protein domains, ProTiler seems specifically designed to do so. In general, ProTiler appears an interesting tool to identify essential protein domains, however it's utility should be more convincingly demonstrated with novel findings. The proteome-wide prediction seems interesting but its utility is not demonstrated and should be better validated.

Major points:

The authors should clearly demonstrate the different utility of ProTiler as compared to CRISPRO and discuss this in the manuscript.

The utility of ProTiler to identify novel essential domains is shown and validated with only one gene (SMARCB1). More experimental data demonstrating that ProTiler can effectively predict and identify essential protein domains in other genes would strengthen the manuscript. To convincingly demonstrate a broad utility of ProTiler authors should identify additional novel protein domains and experimentally validate these.

The proteome-wide CKHS prediction should be truly validated. The proteome-wide CKHS prediction model was trained with the Munoz tiling data set, but it was not demonstrated it can predict the outcome of other tiling screen data. Although it was shown the model has some predictive power (Fig. 4h) authors should demonstrate the correctness of the predicted regions by comparison of the predictions with a ProTiler analysis of a published tiling screen data set. In addition, to demonstrate the utility of the prediction authors should validate a few of the identified novel essential domains from Supplementary Table 5. Ideally, authors could experimentally validate their prediction model by performing a small tiling library screen containing a few genes that were not yet studied with a tiling screen.

Because the domain analysis relies on in-frame indels, one suggestion to improve ProTiler could be to include the prediction of the in-frame frequency by inDelphi to correct the z-scores of the guides.

Specific comments:

- Line 43: It has been demonstrated that indels are non-random but occur in a sequence specific manner and not in a stochastic manner. As the authors mention in line 77 repair outcome probabilities can be predicted from the sequence.
- Line 78: "the model", please specify for easy reading which model.
- Line 104: please explain to the reader how it is determined that these sgRNAs are associated with inactive sgRNAs. Did the authors only use the Doench score for this calculation or also other scores?
- Line 111: What is the cut-off z-score for classifying a region as CKHS or non-CKHS?
- Line 118: Why did the authors use 108 genes from the Munoz data and not all 139 genes? 175 CKHS regions were found in 83 genes, do the remaining genes not contain any functional domains?
- Line 124 typo: 'The borders CKHS'
- CKHS region in N-terminus of SMARCB1 (Fig. 2). ProTiler identified a CKHS region in the N-terminus of SMARCB1. To demonstrate the essentiality of this domain authors expressed a

truncated form a SMARCB1 that could not rescue the endogenous SMARCB1 knockout. Using N-terminal deletion mutants is somewhat drastic to show functionality of a protein domain, there could be many other reasons (e.g. structural defects) why a truncated protein is not functional than just because the deleted domain is essential for function. The authors could demonstrate that subtle in-frame mutations in this domain abolish protein function while the same subtle in-frame mutations in the protein but not in the identified CKHS domain are better tolerated. Why did the authors introduce so many synonymous mutations in the exogenous SMARCB1 constructs while they could simply mutate the PAM (e.g. CGG to CGA)?

- Line 167 typo "Mounz"
- Line 181: type 2 times "inhibition"
- Line 257: To demonstrate the utility of ProTiler the authors could show an example of where ProTiler identified a novel domain in a protein lacking domain information and validate these.
- At several steps throughout the manuscript the results should be better explained in the text, for example lines 258-262 it could be better explained that the two large data sets are from knock-out libraries with only few guides per gene.
- Why does Supplementary Table 2 counts 159 genes while the Munoz paper states 139 genes (p906, line 9)?

Point to point responses to the reviewers' comments

Reviewer #1:

In this paper He et al. present ProTiler, a computational method to analyze tiling data from CRISPR/Cas9 knockout screen aimed at uncovering protein domains. The manuscript is well written and the software well documented. Interestingly the method identified unannotated domains of SMARCB1 that were subsequently validated by the authors. I think this manuscript needs only minor revisions and it could benefit addressing the following points before publication:

1. Line 43: "This is likely because CRISPR/Cas9 introduces small indels that create frameshift or in-frame mutations in a stochastic manner." This contradicts the recent prediction models that can predict alleles after DSBs, one of which is used in this paper. (inDelphi, lines77-79).

In this sentence, the word "stochastic" is misleading. We have modified the sentence into: "This is likely because CRISPR/Cas9 introduces small indels that generally lead to either frameshift or in-frame mutations."

2. Despite a competing method called CRISPRO is mentioned in the introduction no comparison with ProTiler is presented. I think it is important to directly and quantitatively compare the two methods with the Munoz dataset and highlight unique and common features.

We performed a comprehensive analysis to compare ProTiler and CRISPRO, and added the comparative methods and results to the manuscript as follows:

- a) *In Supplementary Fig. 10a, we used the example of SMC2 to show that ProTiler identified broad regions that match Pfam domains well, whereas the CRISPRO prediction includes discontinued segments and false positives. This difference is largely due to the fact that ProTiler includes a robust algorithm to reduce the impacts of inactive sgRNAs and outlier data points.*
- b) *In Supplementary Fig. 10b, we summarized the quantitative measures of sensitivity, specificity, and resolution of the two methods. These results clearly showed higher performance of ProTiler for detecting protein domains.*
- c) *In the discussion section, we added a paragraph to compare the pros and cons of the two methods: "CRISPRO is the first computational tool for mapping the sgRNA functional scores to proteins(Schoonenberg, et al., 2018). Compared to CRISPRO, ProTiler includes an algorithm to reduce the impacts of inactive sgRNAs and outlier data points, which often result in discontinued segments and false positives (Supplementary Fig. 10a). In addition, ProTiler has a robust region calling algorithm to facilitate a de novo analysis. A quantitative comparison of these two methods on the Munoz data showed that ProTiler outperforms CRISPRO in*

sensitivity, specificity, and resolution (Supplementary Fig. 10b). On the other hand, CRISPRO is advantageous in providing an integrative view of genomic, transcriptomic, proteomic, and structural information, and it is suitable for the analysis of well-characterized proteins.”

d) We added the detailed method of comparison in the Methods section.

3. “To ensure robustness against prediction error, we selected the sgRNAs that were predicted to be very likely (top 5%, in-frame probability > 0.60, “high in-frame”) or very unlikely (bottom 5%, in-frame probability < 0.129, “low in-frame”) to create in-frame mutations (Fig. 1b).” I think it would be really helpful to know how this affects the tiling resolution. Could you please create a plot showing the distribution of the distances of consecutive sgRNAs before and after this filtering?

We didn't use inDelphi prediction results to filter the tiling CRISPR signals, because majority of the sgRNAs create a mixture of frameshift and in-frame indels, and only 11.2% of sgRNAs have a predicted in-frame probability greater than 0.5 (Fig. 1b and discussion section). Indeed, we filter out the sgRNAs of weak signals, corresponding to inactive sgRNAs (step1 of the workflow in Fig. 1d). 7,564 out of 28,951 (26.1%) sgRNAs were filtered out in this step. To assess the effect of sgRNA filtering on the tiling resolution, we followed the reviewer's advice to plot the distribution of the distances of consecutive sgRNAs before and after filtering, as shown below:

The distribution shows only slight change and the median distance of consecutive sgRNAs remains the same before and after filtering, so we don't think the filtering will significantly compromise the tiling resolution.

4. “The data points with weaker dropout effects compared to their neighbors are likely to be associated with inactive sgRNAs (Supplementary Fig. 2), thus are removed.” Briefly explain how this filter is implemented/what is the threshold.

Approximately 1/3 of the sgRNAs with a PAM-appended target are inactive, corresponding to weak dropout effects (Rosenbluh, et al., 2017; Xu, et al., 2015). To remove weak dropout signals, each sgRNA data point is compared to its k neighbors to the left and k neighbors to the right. The data point is removed if the signal is weaker than 2/3 of left neighbors and 2/3 of right neighbors. We set $k=5$, corresponding to an average window span of ~30 AAs, the size of the smallest protein domain module. We have added this detailed description to the Methods section in the manuscript.

5. “We adjusted their values based on the mean and variation of the surrounding signals.” See the previous point. Is this a smoothing?

The purpose of this step is to reduce the impact of “outliers” on CKHS region calling. This is not exactly a smoothing procedure, since majority of the data points are unchanged. To adjust the outliers, we estimated the variation of noise for each protein, by applying Median Absolute Deviation (MAD) on the differences between consecutive sgRNA signal. For each data point x , we compare it to the median value of its neighbors within a sliding window of size 11 (approximately the size of the smallest known protein domain module, see reply to review #1, point 4). If x is larger than the median value by more than twice of MAD, x is marked to be an outlier and is adjusted to be $\text{median}+2\text{MAD}$. The outliers below the median values are detected in a similar way and are adjusted to $\text{median}-2*\text{MAD}$.*

We have added this detailed description to the Methods section in the manuscript. In the Result section, we added the sentence “To reduce the impacts of the “outliers” ...” to clarify the rationale of this step.

6. “Finally, an iterative algorithm classifies the regions into CKHS and non-CKHS categories.” See 4) and 5). Add a very short sentence to explain what is the key idea of this algorithm.

This is a “ k -means-like” algorithm ($k=2$) for CKHS region calling. Initially, the region with the most significant drop-out score is assigned to the CKHS group, and the rest of the regions are assigned to the non-CKHS group. In each iteration, the regions are assigned to one of the groups based on the distance to the weighted average score of each group computed in the previous iteration. The pseudo-code of the algorithm is added in the Methods section of the manuscript. We also added a short sentence in the Results section to briefly describe the algorithm.

7. “Among 108 essential proteins in the Munoz data, ProTiler identified 175 CKHS regions in 83 proteins. 82.3% of these regions overlapped with Pfam annotated protein domains” and “At the amino acid (AA) level, 64.2% of the AAs in the CKHS regions are within Pfam domains, compared to 30.0% for non-CKHS regions”. Please report also the opposite i.e. how many annotated Pfam protein domains in the analyzed genes were missed by ProTiler.

We have added the summary statistics: "In sensitivity, 207 out of 277 (74.7%) Pfam domains in the 83 proteins were identified.". Of note, the annotated Pfam domains are not necessarily essential in the experiments. The percentage of missed domains is not a true estimation of false negatives.

8. "The borders CKHS regions were enriched within 20 AAs from domain boundaries and slightly outside the domains". Please provide a quantitative measure for the enrichment. Also, a simple measure to assess performance would be the F1-score. In fact, this problem can be thought as a binary classification problem: each amino acid in the protein is a domain (1) or not (0).

We have added p-values to measure the statistical enrichment of the boundary overlap: " The left and right borders of CKHS regions were enriched within 20 AAs from domain boundaries (left: $p = 5.87e-08$; right: $p = 9.36e-06$, Fisher exact test)".

We have computed F1-score as a measurement of performance, and used random permutation to compute the distribution of F1-scores by random expectation. We added the results to the Result section and Supplementary Fig. 4. The F1-score has also been used for the comparison with CRISPRO.

Reviewer #2:

In this study He et al. describe how a tiling sgRNA library can be applied to identify essential protein domains. They developed an algorithm named ProTiler to map hypersensitive CRISPR knockout regions based on the severity of sgRNA dropout from a tiling screen. They applied ProTiler on a published tiling screen dataset from Munoz et al. and identified hypersensitive regions that largely overlap with annotated Pfam domains as well as known drug target domains in proteins from this dataset. They also identified a new domain in SMARCB1 and predicted hypersensitive regions in a proteome-wide scale based on protein features. Previously Schoonenberg et al. developed a similar algorithm named CRISPRO for which the focus appears more on the visualization of essential domains in the protein structure and on prediction for the efficiency of guides. However, Schoonenberg et al. also describe the capacity of CRISPRO to highlight important protein regions and help prioritize protein regions of interest for chemical biology. Although CRISPRO also enables to detect essential protein domains, ProTiler seems specifically designed to do so. In general, ProTiler appears an interesting tool to identify essential protein domains, however it's utility should be more convincingly demonstrated with novel findings. The proteome-wide prediction seems interesting but its utility is not demonstrated and should be better validated.

We thank reviewer for the thoughtful review of our manuscript. To address the reviewer's concerns, we have i) performed comprehensive analysis to compare CRISPRO and ProTiler for domain identification and discussed their pros and cons for different utilities. ii) performed additional experiment to validate the function of identified N-terminal domain of SMARCB1. iii) demonstrated the utility of proteome-wide prediction through other tiling screen data and two newly reported novel domains. Following are the detailed responses to the reviewer's comments.

Major points:

1. The authors should clearly demonstrate the different utility of ProTiler as compared to CRISPRO and discuss this in the manuscript.

We performed a comprehensive analysis to compare ProTiler and CRISPRO, and added the comparative methods and results to the manuscript. Our results clearly showed that ProTiler outperforms CRISPRO for detecting protein domains. For details, please refer to the reply to Reviewer #1, point 2.

2. The utility of ProTiler to identify novel essential domains is shown and validated with only one gene (SMARCB1). More experimental data demonstrating that ProTiler can effectively predict and identify essential protein domains in other genes would strengthen the manuscript. To convincingly demonstrate a broad utility of ProTiler authors should identify additional novel protein domains and experimentally validate these.

We agree to the reviewer that more examples of novel protein domains will be helpful to demonstrate a broad utility of ProTiler. Since this study is mainly focused on the data analysis, we sought to address this question from two perspectives. First, we have strengthened the evidence to show the essentiality of a novel protein domain in SMARCB1 by additional validation experiments (see reply to Reviewer #2, point 11 for details). Second, we show that some examples of novel protein domains (AA 86-99 in YAP1, AA 58-184 in TTK) have been reported and validated in recent publications but were not annotated previously. Taking these lines of evidence together, we show that ProTiler is not only applicable to a single validated example, but has broader application for novel domain discovery. We hope these explanations regarding the utility of ProTiler are acceptable.

For the future work, we do plan to study additional novel domains experimentally, dependent on the biological and clinical significance, expert knowledge of biological system, availability of a potent antibody and the size of exogenous proteins.

3. The proteome-wide CKHS prediction should be truly validated. The proteome-wide CKHS prediction model was trained with the Munoz tiling data set, but it was not demonstrated it can predict the outcome of other tiling screen data. Although it was shown the model has some predictive power (Fig. 4h) authors should demonstrate the correctness of the predicted regions by comparison of the predictions with a ProTiler analysis of a published tiling screen data set. In addition, to demonstrate the utility of the prediction authors should validate a few of the identified novel essential domains from Supplementary Table 5. Ideally, authors could experimentally validate their prediction model by performing a small tiling library screen containing a few genes that were not yet studied with a tiling screen.

First, to check if the SVM model can predict the outcome of other tiling screen data, we applied ProTiler to an independent tiling screen dataset that includes two genes, ZBTB7A and MYB (Schoonenberg, et al., 2018), and compared the results to the SVM prediction. We observed high consistency among the SVM predictions, ProTiler identified CKHS regions, and the Pfam annotated protein domains (Supplementary Fig. 7), which implies the predictive power of our model beyond the scope of Munoz data. We have added these results to the revised manuscript.

Second, we try to explore the utility of the SVM prediction for novel domain discovery. Since this study is mainly focused on the data analysis, we sought to answer this question based on recently reported novel domains. We compared the predicted regions to two recent reports of new functional domains: a non-catalytic “FLOS” domain in methyltransferase SETD1A (Hoshii, et al., 2018), and an “eMIC” domain in splicing factor SRRM4 (Torres-Mendez, et al., 2019). In both proteins, the SVM model correctly predicted the newly identified domains (Supplementary Fig. 8). We have added these results to the revised manuscript.

Third, we agree that additional tiling screens will be useful for further exploration of new domain functions, especially for uncharacterized proteins. Since the studies

presented in the paper is mainly focused on the data analysis, we consider the additional screens as follow-up studies of this project, which are discussed in the revised version of the manuscript.

Finally, we'd like to point out that the SVM prediction is imperfect (AUC between 0.7-0.8). Although it provides insights into novel domain discovery, additional proteomic data and functional validations are needed. On the other hand, an immediate application of the prediction is to improve the sgRNA design for genome-scale CRISPR screen, as shown in Fig 4h. We have added these discussions to the revised manuscript.

4. Because the domain analysis relies on in-frame indels, one suggestion to improve ProTiler could be to include the prediction of the in-frame frequency by inDelphi to correct the z-scores of the guides.

It is a good idea to include in-frame frequency predictions to improve ProTiler. We have tried to filter the sgRNAs based on prediction of the in-frame frequency. However, since i) computational predictions are subject to false positive and false negatives; ii) only 11.2% of sgRNAs have a predicted in-frame probability greater than 0.5 (Fig. 1b), a simple filtering strategy does not improve the analysis, and indeed compromises the resolution. We believe a more complex computational framework is needed to improve ProTiler using predicted in-frame frequency, which is one of our future direction. We have discussed this issue in the revised manuscript.

Specific comments:

5. Line 43: It has been demonstrated that indels are non-random but occur in a sequence specific manner and not in a stochastic manner. As the authors mention in line 77 repair outcome probabilities can be predicted from the sequence.

Yes, in this sentence, the word "stochastic" is misleading. We have modified the sentence into: "This is likely because CRISPR/Cas9 introduces small indels that generally lead to either frameshift or in-frame mutations."

6. Line 78: "the model", please specify for easy reading which model.

We have changed it to "the in-frame indel model" in the revised manuscript. References are given to Fig. 1a.

7. Line 104: please explain to the reader how it is determined that these sgRNAs are associated with inactive sgRNAs. Did the authors only use the Doench score for this calculation or also other scores?

We used two models to assess the sgRNA activities: i) the Doench model (Doench, et al., 2016); ii) the SSC model (Xu, et al., 2015). The results are shown in Supplemental Fig. 2. Both results show that the filtered sgRNAs are associated with low activity scores.

8. Line 111: What is the cut-off z-score for classifying a region as CKHS or non-CKHS?

Since different proteins have different essentiality in quantity, it is not applicable to use a fixed cut-off to classify CKHS and non-CKHS region. We used a “k-mean-like” algorithm (k=2) for CKHS region calling (see reply to Reviewer #1, point 6, for details). The pseudo-code of the algorithm is added in the Methods section of the manuscript. We also added a short sentence in the Results section to briefly describe the algorithm.

9. Line 118: Why did the authors use 108 genes from the Munoz data and not all 139 genes? 175 CKHS regions were found in 83 genes, do the remaining genes not contain any functional domains?

The 108 genes were selected as functional essential gene in at least one of the three cell lines in the screen. For genes that are not essential in any of the cell lines, the information is inadequate for CKHS region calling. The details of essential gene identification are described in the method section.

We examined the genes without an identified CKHS region. We found these proteins are associated with relatively smaller size, and often harbor a single domain that covers majority of the amino acids (Supplementary Fig. 3). In these cases, one can expect that a small deletion in most of the AA positions can significantly impact the protein function. Since ProTiler is focused on the detection of protein regions that show stronger knockout effect compared to other regions of the same protein, the CKHS regions were not called for these proteins. This observation also suggest that the tiling screen is more useful for the studies of larger proteins with multiple domains. We have added the description to the revised manuscript for clarification.

10. Line 124 typo: ‘The borders CKHS’

We have fixed the typo to “The left and right borders of CKHS” in the revised manuscript.

11. CKHS region in N-terminus of SMARCB1 (Fig. 2). ProTiler identified a CKHS region in the N-terminus of SMARCB1. To demonstrate the essentiality of this domain authors expressed a truncated form a SMARCB1 that could not rescue the endogenous SMARCB1 knockout. Using N-terminal deletion mutants is somewhat drastic to show functionality of a protein domain, there could be many other reasons

(e.g. structural defects) why a truncated protein is not functional than just because the deleted domain is essential for function. The authors could demonstrate that subtle in-frame mutations in this domain abolish protein function while the same subtle in-frame mutations in the protein but not in the identified CKHS domain are better tolerated. Why did the authors introduce so many synonymous mutations in the exogenous SMARCB1 constructs while they could simply mutate the PAM (e.g. CGG to CGA)?

We have followed the reviewer's advice to test the effect of small deletions of 3 AAs in the identified domain ($\Delta 29-31$), with a deletion of AAs outside the domain ($\Delta 150-152$) as control. Our results clearly show that exogeneous expression of $\Delta 150-152$ SMARCB1 protein can rescue the phenotype, whereas the expression of $\Delta 29-31$ protein fail to rescue (Fig. 2f). These lines of evidence further support the essentiality of the N-terminus domain.

The idea of introducing synonymous mutations is to ensure that the knockouts with sgRNAs targeting the mutated sites (sg1, sg2, sg3 in Fig. 2) would abolish only the endogenous SMARCB1 without compromising the function of the ATP-binding domain in the exogenous proteins. This facilitates the rescue experiments. We agree with the reviewer that mutating PAM could be a simpler way. However, this strategy largely depends on whether certain PAM belongs to one codon or not. In our case, two PAMs (GGG, AGG) for sg1 and sg3 cover two codons, if we mutate them, there is a likelihood of introducing nonsynonymous mutations to the ATP-binding domain, adding confounding factors to the validation of N-terminus domain.

12. Line 167 typo "Mounz"

Typo fixed.

13. Line 181: type 2 times "inhibition"

Typo fixed.

14. Line 257: To demonstrate the utility of ProTiler the authors could show an example of where ProTiler identified a novel domain in a protein lacking domain information and validate these.

This question is related to point 3. We have added the results to show that our predictions are highly consistent with two recently identified novel domains in SETD1A and SRRM4.

15. At several steps throughout the manuscript the results should be better explained in the text, for example lines 258-262 it could be better explained that the two large data sets are from knock-out libraries with only few guides per gene.

We have added information to better explain the libraries, as follow: "..., using two large-scale CRISPR/Cas9 screen datasets based on the GeCKO-v2 and Avana libraries, respectively, in which 4-6 sgRNAs were designed for each gene."

16. Why does Supplementary Table 2 counts 159 genes while the Munoz paper states 139 genes (p906, line 9)?

We have carefully examined Munoz data published in the original paper. There are exactly 159 genes. It seems the number of 139 is a typo in the original paper.

References

- Doench, J.G., *et al.* Optimized sgRNA design to maximize activity and minimize off-target effects of CRISPR-Cas9. *Nat Biotechnol* 2016;34(2):184-191.
- Hoshii, T., *et al.* A Non-catalytic Function of SETD1A Regulates Cyclin K and the DNA Damage Response. *Cell* 2018;172(5):1007-1021 e1017.
- Rosenbluh, J., *et al.* Complementary information derived from CRISPR Cas9 mediated gene deletion and suppression. *Nat Commun* 2017;8.
- Schoonenberg, V.A.C., *et al.* CRISPRO: identification of functional protein coding sequences based on genome editing dense mutagenesis. *Genome Biol* 2018;19(1):169.
- Torres-Mendez, A., *et al.* A novel protein domain in an ancestral splicing factor drove the evolution of neural microexons. *Nat Ecol Evol* 2019;3(4):691-701.
- Xu, H., *et al.* Sequence determinants of improved CRISPR sgRNA design. *Genome Res* 2015;25(8):1147-1157.

Reviewers' Comments:

Reviewer #1:

Remarks to the Author:

I think the authors did an excellent job in addressing all my concerns (and also the ones from the other Reviewer) and I look forward to seeing this manuscript online.

Reviewer #2:

Remarks to the Author:

The authors have responded sufficiently to the reviewers comment and I have no further remarks.

Point-by-point responses to reviewers' and editor's comments

Reviewer #1 (Remarks to the Author):

I think the authors did an excellent job in addressing all my concerns (and also the ones from the other Reviewer) and I look forward to seeing this manuscript online.

We thank the reviewer for reviewing our manuscript.

Reviewer #2 (Remarks to the Author):

The authors have responded sufficiently to the reviewers' comment and I have no further remarks.

We thank the reviewer for reviewing our manuscript.